# A Phase 2a Randomized, Double-Blind, Dose-Optimizing Study to Evaluate the Immunogenicity and Safety of a Bivalent DNA Vaccine for Hemorrhagic Fever with Renal Syndrome Delivered by Intramuscular Electroporation

**DOI:** 10.3390/vaccines8030377

**Published:** 2020-07-11

**Authors:** Jay Hooper, K. M. Paolino, K. Mills, S. Kwilas, M. Josleyn, M. Cohen, B. Somerville, M. Wisniewski, S. Norris, B. Hill, M. Sanchez-Lockhart, D. Hannaman, C. S. Schmaljohn

**Affiliations:** 1Virology Division, United States Army Medical Research Institute of Infectious Diseases, Fort Detrick, MD 21702, USA; steven.a.kwilas.ctr@mail.mil (S.K.); matthew.d.josleyn.civ@mail.mil (M.J.); melanie.cohen@nih.gov (M.C.); brandon.c.somerville.ctr@mail.mil (B.S.); mlw53@cornell.edu (M.W.); sarah.l.norris2.civ@mail.mil (S.N.); brenna.hill@nih.gov (B.H.); mariano.sanchez-lockhart.ctr@mail.mil (M.S.-L.); connie.schmaljohn@nih.gov (C.S.S.); 2Clinical Trials Center, Walter Reed Army Institute of Research, Silver Spring, MD 20910, USA; paolinok@upstate.edu (K.M.P.); Kristin.mills@optum.optumcare.com (K.M.); 3Ichor Medical Systems, Inc., San Diego, CA 92121, USA; dhannaman@ichorms.com

**Keywords:** hantavirus, hemorrhagic fever with renal syndrome, DNA vaccine, electroporation

## Abstract

Hantaan virus (HTNV) and Puumala virus (PUUV) are pathogenic hantaviruses found in Asia and Europe, respectively. DNA vaccines targeting the envelope glycoproteins of these viruses have been constructed and found to elicit neutralizing antibodies when delivered to humans by various technologies including intramuscular electroporation. Here, we report findings from a Phase 2a clinical trial of a combined HTNV/PUUV DNA vaccine delivered at varying doses and administration schedules using the Ichor Medical Systems TriGrid intramuscular electroporation delivery technology. The study was designed to characterize the effects of DNA vaccine dose and number of administrations on the frequency and magnitude of immunological response. Subjects (*n* = 120) were divided into four cohorts. Cohorts 1 and 2 received a dose of 2 mg of DNA (1 mg per plasmid), and cohorts 3 and 4 received a dose of 1 mg of DNA (0.5 mg per plasmid) each vaccination. Each of the four cohorts received a series of four administrations (days 0, 28, 56 and 168). For cohorts 1 and 3, the DNA vaccine candidate was delivered at each of the four administrations. For cohorts 2 and 4, in order to maintain blinding, subjects received the DNA vaccine on days 0, 56 and 168, but on day 28 received only the phosphate buffered saline vehicle rather the DNA vaccine. Sera were collected on days 0, 28, 56, 84, 140, 168, 196, 252 and 365 and evaluated for the presence of neutralizing antibodies by PUUV and HTNV pseudovirion neutralization assays (PsVNAs). Day 84 was also evaluated by a plaque reduction neutralization test (PRNT). Overall the PsVNA50 geometric mean titers (GMTs) and seropositivity rates among cohorts were similar. Cohort 3 exhibited the highest frequency of subjects that became seropositive to both PUUV and HTNV after vaccination, the highest peak GMT against both viruses, and the highest median titers against both viruses.

## 1. Introduction

Most cases of hemorrhagic fever with renal syndrome (HFRS) are caused by infection with Hantaan (HTNV) or Seoul (SEOV) viruses in Asia and by Puumala (PUUV) or Dobrava (DOBV) viruses in Scandinavian countries and other parts of Europe. Pathogenic hantaviruses are carried by persistently infected rodents and are transmitted to humans by ingestion or inhalation of rodent excreta, or occasionally by bite. HFRS is a significant health threat in endemic areas with thousands of hospitalized cases reported each year in China [1] and several hundred to thousands of HFRS cases occurring annually in Europe and Russia [2,3,4].

There are currently no FDA-licensed vaccines for HFRS and treatment generally consists of supportive care. Ribavirin has been used off-label to treat HFRS, and is reported to reduce mortality when given early, but not late, in the disease course [5,6]. In addition to having limited efficacy, ribavirin causes a reversible hemolytic anemia and numerous IV injections of the drug must be given over a 7-day period [6]. Toward the goal of licensing a safe and effective vaccine for HFRS, we have developed a two-plasmid DNA vaccine expressing the envelope glycoprotein (Gn and Gc) genes of HTNV or PUUV. Neutralizing antibodies to Gn and Gc have been shown to be sufficient to confer protective immunity against infection in hamsters, and are believed to be a key component of the protective immune response in humans as well [7,8,9,10,11].

We previously reported two Phase 1 clinical studies in which both the HTNV and PUUV plasmid DNA vaccines were found to be safe and immunogenic when delivered by a gene gun [12] or by intramuscular electroporation (IM-EP) [13]. In addition, the IM-EP study demonstrated that immune responses to both HTNV and PUUV could be elicited with a 1:1 mixture of the plasmid DNAs. Here we report the results of a Phase 2a, randomized, dose-optimizing trial of the mixed HTNV and PUUV DNA vaccines delivered by IM-EP. We evaluated safety and compared neutralizing antibodies elicited by the DNA vaccines at two doses and two schedules. This study is intended to inform decisions on selecting an effective dose and schedule for further larger scale hantavirus vaccine studies.

## 2. Materials and Methods 

### 2.1. Vaccines

The HTNV DNA vaccine plasmid, pWRG/HTN-M (co), was constructed by cloning cDNA representing the HTNV M segment open reading frame (ORF), which encodes Gn and Gc, into the *Not*I and *Bgl*II-restriction sites of pWRG7077 [14] as described previously [10]. Optimization (GeneArt, Regensburg, Bavaria, DE, Germany) of the M ORF was performed to modify the codons for *Homo sapiens* usage, and to remove gene elements known to interfere with mammalian expression and mRNA stability. This optimized DNA eliminates the requirement for an untranslated extraneous sequence upstream of the ORF to be included in the plasmid as described previously [10], and also eliminates interference issues that occurred with the non-optimized DNA vaccine, pWRG/HTN-M(x), in animals and humans [13,15]. The PUUV DNA vaccine plasmid, pWRG/PUU-M(s2), was constructed similarly using cDNA that was engineered as a consensus sequence of several PUUV isolates, and optimized for codon usage (*Homo sapiens*) and mRNA stability (GeneArt, Regensburg, Bavaria, DE, Germany) [16]. The HTNV and PUUV DNA vaccines were produced under current good manufacturing practices (cGMP) by Althea Technologies, Inc. (San Diego, CA, USA). The DNA was formulated at 2 mg/mL in phosphate buffered saline (PBS) (Thermo Fisher Scientific, Waltham, MA, USA). The potency of the DNA vaccines was measured by evaluating expression of the hantavirus glycoproteins using a standardized flow-cytometry-based in vitro potency assay performed essentially as described previously [12,17]. On the day of vaccination, a simple 1:1 mixture of both vaccines was prepared by combining equal volumes of HTNV and PUUV DNA vaccines in a separate vial.

### 2.2. TriGrid Intramuscular Delivery System (TDS-IM)

The clinical use of the TDS-IM EP delivery device (Ichor Medical Systems, Inc., San Diego, CA, USA) has been described [13]. The total duration of the electrical stimulation is 40 milliseconds applied over a 400 millisecond interval (a 10% duty cycle).

### 2.3. Clinical Study Subjects

Healthy adult volunteers, male and female, between the ages of 18 and 49 (inclusive) were recruited through the Walter Reed Army Institute of Research (WRAIR) Clinical Trials Center, Silver Spring, Maryland. All recruiting and consent methods and materials were compliant with current good clinical practice (GCP) guidelines and approved by the Walter Reed Army Institute of Research (WRAIR) Institutional Review Board (IRB). All study procedures took place at this site. 

Data obtained from all subjects that received at least one vaccination were included in the safety statistical analysis. Exclusion criteria included pregnant or lactating females, a history of severe reactions to any vaccination or a history of severe allergic reactions, acute or chronic medical or psychiatric conditions, receipt of another vaccine or IND product within 30 days of the planned first dose, receipt of blood products within 120 days prior to enrollment, immunosuppressive or immunodeficient conditions and chronic use of immunosuppressive drugs other than inhaled and topical steroids. To ensure that vaccine injections with the TDS-IM were administered intramuscularly in all subjects, individuals with a skinfold measurement of the cutaneous and subcutaneous tissue for all eligible injection sites (deltoid region) exceeding 40 mm were also excluded. To exclude persons with possible prior exposure to a hantavirus, serum samples from all subjects were screened at a 1:20 dilution for pre-existing antibodies to HTNV and PUUV using HTNV and PUUV pseudovirion neutralization assays (PsVNAs). Only seronegative subjects were enrolled in the study.

### 2.4. Clinical Study Design Overview

The single-center study was sponsored by the Office of the Surgeon General, Department of the Army, under IND 14828. The double-blinded study included four experimental cohorts, which varied in either the DNA vaccine dose or administration schedule. Following consent and successful screening, each subject was randomized using a simple randomization method to allocate subjects into one of the four experimental groups. Subject ID numbers were randomly preassigned to one of the four groups, according to a list made by the study statistician prior to screening and enrollment. Subjects were assigned ID numbers sequentially as they were enrolled into the study. The final (up to 12) eligible subjects were allocated as alternates to replace any original subjects who failed to complete all 3 scheduled primary injections and the day 70 follow-up visit. The alternates were used in order of enrollment to fill any openings in the groups as they arose. The four cohorts consisted of 30 subjects each for a total of 120 subjects. Subject demographics are reported in Table 1. The cohorts were split, so that 60 individuals received a 1.0 mg dose in 1 mL of normal sterile saline (0.9% NaCl, Injection, USP, Hospira, Lake Forest, IL, USA), and the other 60 received a 2.0 mg dose in 1 mL of sterile saline. During the initial administration sequence, every subject received a total of 3 injections, on days 0, 28 and 56. Half of each of these cohorts received two DNA vaccine injections at day 0 and 56 and a sterile saline placebo on day 28, while the other half received 3 DNA vaccine injections at day 0, 28 and 56. All cohorts received a booster dose at Day 168. All doses were administered with the TDS-IM device to the deltoid region on alternating arms beginning with the left. All subjects completing the study were followed until at least day 252. Subjects that were positive for neutralizing antibodies on day 252 were requested to return for a 12 month follow-up visit. Subjects completed post-injection memory aids for 7 days after each injection.

### 2.5. Safety Assessments

Safety was assessed by evaluating the reactogenicity during specified periods of the study. Local and systemic reactions were reviewed with subjects. The following endpoints were evaluated: (1) the nature, frequency and severity of solicited adverse events (AEs) occurring from the time of each injection through 14 days following the procedure; (2) the nature, frequency and severity of unsolicited AEs from the time of the first injection through 28 days following the final injection and (3) the nature, frequency and severity of AEs from the time of the first injection through the end of the study. The solicited AEs for this study included: local findings at the site of injection (redness, swelling, bruising or pain), fever, myalgia, muscle contractions, fatigue, headache, lymphadenopathy, axillary pain or discomfort and tachypnea. Inherent in this assessment were the medical and clinical considerations of all information surrounding the event including any medical intervention required. Each event was assigned one of the following categories: Grade 1 (mild, does not interfere with routine activities); Grade 2 (moderate, interferes with routine activities); Grade 3 (severe, unable to perform routine activities) and Grade 4 (hospitalization or ER visit for potentially life-threatening event). Any Grade 4 AE was reported as a serious adverse event (SAE). Laboratory AEs and abnormalities in subject vital signs were also assessed and graded using pre-specified normal ranges within the study protocol. Safety labs included sodium, potassium glucose, blood urea nitrogen (BUN), creatinine, alkaline phosphatase, aspartate aminotransferase (AST), alanine aminotransferase (ALT), complete blood count with differential and urinalysis.

### 2.6. Pseudovirion Neutralization Assay (PsVNA)

To assess immunogenicity, all specimens were evaluated for the presence of neutralizing antibodies using a pseudovirion neutralization assay (PsvNA) [18,19]. The USAMRIID PsVNA is an investigational assay that has been used previously in nonclinical hantavirus research [19,20]. The PsVNA utilizes engineered vesicular stomatitis virus (VSV) that expresses a luciferase reporter gene in the place of the virus G envelope glycoprotein genes [21]. Briefly, pseudovirions (PsV) were purified and quantified as described previously [19]. To perform the neutralization assay, HTNV or PUUV PsVs (4000 focus forming units) were combined with serum (1:20–1,562,500 dilution range) in the presence of a human complement (5%; Sigma, St. Louis, MO, USA) and incubated overnight at 2–8 °C. The PsV plus serum mixture were then added to Vero-76 cell monolayers in clear bottom black-walled 96-well microtiter plates. The plates were incubated 18–24 h and then media removed, lysis luciferase reagent (Promega, Madison, WI, USA) added and flash luminescence data acquired using a luminometer (Tecan M200 Pro microplate reader, Mannedorf, CH, Switzerland). If sera contain antibodies that prevent the PsV from attaching to and/or entering cells, then the reporter activity is neutralized. Neutralization titers are interpolated from 4-parameter curves using GraphPad Prism (GraphPad, San Diego, CA, USA). The reciprocal of the interpolated dilution that results in a 50% decrease, or 80% decrease in luciferase activity is the PsVNA50, or PsVNA80 titer, respectively.

### 2.7. Plaque Reduction Neutralization Test (PRNT)

Day 84 specimens were evaluated for neutralizing antibodies by plaque reduction neutralization test (PRNT) as described previously [12,22,23]. For the HTNV and PUUV PRNT, monolayers were fixed at 7 and 10 days after infection, respectively, by the addition of 2 mL of 10% formalin/well followed by 5–16 h incubation at room temperature. The agarose overlay was removed with a flat spatula and the plaques were visualized by immunostaining using horseradish peroxidase-conjugated monoclonal antibody MAb-3d7 followed by True Blue peroxidase substrate (KPL, Gaithersburg, MD, USA). Blinded serum samples from subjects were either screened at a 1:20 dilution for HTNV or PUUV neutralizing activity and those positive were then retested in duplicate to determine endpoint HTNV and PUUV PRNT50 titers; or alternatively, full endpoint neutralizing antibody titers against both HTNV and PUUV were determined in duplicate without previous prescreening of the blinded samples.

### 2.8. HLA Typing

Following the manufacturer’s protocol, DNA was extracted from PBMCs using DNAzol^TM^ reagent (Invitrogen, Waltham, MA, USA). DNA quantity and quality was verified using a NanoDrop™ 2000 Spectrophotometer (Thermo Fisher Scientific, Waltham, MA, USA). DNA was amplified using different pairs of primers specific for each HLA allele using the HOLOTYPE HLA™ (Omixon, Cambridge, MA, USA). Amplicons were clean using Agencourt AMPure XP beads (Beckman Coulter Life Sciences, Brea, CA, USA) and cleaned product assessed by Agilent 2100 Bioanalyzer (Agilent, Santa Clara, CA, USA). Correct size product range from 257 to 651 bp, depending on the HLA amplicon generated. KAPA Library Preparation Kit Illumina^®^ Platforms (KAPA Biosystems, Wilmington, MA, USA) was used to prepare the sequence library, and a KAPA Library Quantification Kit Illumina^®^ Platforms (KAPA Biosystems, Wilmington, MA, USA) for quantification. Sequencing libraries were pooled and sequenced using 2 × 250 cycles on a MiSeq System (Illumina^®^ Inc., San Diego, CA, USA). HLA typing was done using the HLA TwinTM (Omixon, Cambridge, MA, USA).

The titer response outcome variable was defined as a binary variable (responder or nonresponder). A responder was defined as those with PsVNA50 titers greater than 20. A nonresponder was defined as those with PsVNA50 titers less than or equal to 20.

Allele effects on the titer response were tested using a logistic regression model. More than half of all subjects were missing any alleles for loci F, Y, DRB2, DRB4, DRB5, DRB7 and DRB8; therefore, those loci were not analyzed. For loci with less than 50% of subjects missing alleles, frequencies of two-digit and four-digit alleles were calculated. Those two-digit and four-digit alleles that were present in at least 10% of subjects were chosen for further analysis. Alleles of interest were coded as 1 (target present) or 0 (other allele present). Subjects with no allele present on the corresponding locus were excluded; however, the frequencies of response/nonresponse are also presented. Each subject contributed up to two alleles per locus. Each allele variable at a locus was included in a separate logistic regression model to determine its association with titer response against all other alleles at the same locus combined. Multiple testing in alleles was corrected using false discovery rate (FDR). These adjusted *p*-values are represented as q-values with a cutoff for significance at alpha = 0.05. Analysis was performed using SAS Version 9.4 (SAS Institute, Cary, NC, USA).

### 2.9. Statistical Methods

Descriptive analysis of safety and reactogenicity outcomes included all subjects who met the eligibility criteria, received at least one vaccination, and for whom safety data were available. Summary tables were created in which incidence, intensity and the relationship to use the investigational product of individual solicited signs, symptoms and other events were delineated by the study cohort, severity, sex and overall. Unsolicited AEs and SAEs were analyzed in a similar fashion. For hematology and serum chemistry tests, any clinically significant change from the baseline value was identified. The median, interquartile range and normal values for each of the laboratory values (as determined by the contract laboratory) were reported for each treatment cohort for each specimen collection point. The primary analysis variable was the proportion of seropositive subjects (PsVNA50 and/or PRNT50 ≥ 20) at each scheduled time point for which blood samples were taken and duration of seropositivity. Geometric mean peak PsVNA50 and PRNT50 titers were also determined for specified timepoints. Values below each assay’s limit of detection (20 for all assays) were set to 14.14 (20/√2) for analysis. Due to the geometric progression of the assay results, log10 transformations were applied to approximate normality. For all methods of comparisons, transformed data were used. To assess agreement between PsVNA and PRNT values, the day 84 results of the two assays were analyzed using four different methods: Pearson product moment correlation, mountain plot, Bland–Altman plot and Deming regression analysis.

### 2.10. Data Quality Assurance

The WRAIR Clinical Trials Center conducts studies according to procedures that incorporate the ethical principles of the GCP guidelines. To ensure compliance with these procedures and to assess the adequacy of quality control procedures, the WRAIR Quality Office performed audits of the study site on behalf of the USAMRIID Quality Assurance and Regulatory Compliance Office (QARCO). Quality assurance responsibilities included visits at the initiation of the study, during the study at appropriate intervals, and after the last subject had completed the study. The WRAIR Quality Office performed the audits independently of the study monitors.

## 3. Results

### 3.1. Clinical Subject Population

Prior to enrollment, 178 individuals were screened at a 1:20 serum dilution for pre-existing anti-hantavirus neutralizing antibodies by HTNV and PUUV PsVNA. Sixteen individuals were positive in the HTNV PsVNA screen (i.e., >48% neutralization at a 1:20 dilution) and seven of those were confirmed to be positive (i.e., PsVNA50 titer ≥ 20). Pre-existing positive HTNV PsVNA50 titers ranged from 22 to 1059. Four individuals were positive in the PUUV PsVNA and 1 of those was confirmed to have titer ≥20 (i.e., PUUV PsVNA50 = 40). That same individual was also positive in the HTNV PsVNA. Thus, seven of 178 (3.93%) of screened subjects were excluded from enrollment based on preexisting seropositivity in the PsVNA.

The Phase 2a study enrolled four randomized cohorts of 30 subjects each, along with 10 alternates, for a total of 130 subjects (Figure 1). Subjects were enrolled at 9 time points between 7/21/2014 and 12/8/2015. The subjects enrolled included 67 males and 63 females between the ages of 18–49. Races enrolled included White, African American, Asian and other. The ethnicities included 11 Hispanic/Latino subjects (Table 1). 

### 3.2. Vaccination and Safety Assessment

All subjects who received at least a single dose of vaccine were evaluated for safety. No SAEs related to the vaccine or study-related procedures were observed. The most common local solicited AE was pain at the site of injection, which was reported by 125 of the 130 subjects, with 18 subjects experiencing injection site erythema and 9 subjects also showing bruising at the site of injection (Table 2). The next most common solicited AEs were fatigue (38 subjects), headache (31 subjects) and myalgia, described as muscle aches, (23 subjects; Table 2). All study-related adverse events were graded as mild, moderate or severe. There was one unanticipated event that occurred during vaccine administration. Specifically, the TDS-IM device detected a fault in the electrode/needle insertion prior to administration of the injection. The detected error reflected an incorrect needle insertion depth setting issue and appeared to have occurred due to contact of the electrodes/injection needle with the subject’s periosteum. Although this was unexpected, it did not result in a SAE and the subject reported only grade 2 injection site pain and swelling consistent with reports from other recipients; there were no other medical complications.

### 3.3. Neutralizing Antibody Response as Measured by PsVNA

Of the 130 enrolled subjects, 120 met the conditions for the efficacy evaluable population. The efficacy evaluable population included subjects who received all three of the first three vaccinations (days 0, 28 and 56) and attended at least one scheduled study visit subsequent to receiving the third vaccination on day 56. Individual PsVNA50 titers for each cohort were determined. After data-lock individual PsVNA50 titers for HTNV and PUUV for each cohort were plotted (Figure 2).

Overall group specific neutralizing responses (Figure 3) to PUUV measured for cohorts 1–4, respectively, were 86.7%, 80.0%, 83.3% and 73.3%; and to HTNV were 80.0%, 90.0%, 90.0% and 83.3%. None of the subjects were seropositive for PUUV or HTNV neutralizing antibodies on day 0, confirming that all enrolled subjects were negative in the PsVNA prior to the first vaccination. One month (day 28) after the first vaccination a small number (between 1 and 5 per cohort) of subjects had developed neutralizing antibodies against PUUV and/or HTNV. After the second vaccination (day 56) there was a significant (*p*-value < 0.05) difference between cohorts that had or had not received the day 28 boost. This was true for both HTNV and PUUV neutralizing responses (Figure 3). The only other statistically significant difference between the seropositivity rates for the different cohorts was for the PUUV PsVNA50 on day 196 where cohort 3 was significantly higher than cohort 4. There was a peak of seroconversion approximately 1 month after the first series of vaccinations (day 84) and then another peak approximately 1 month after the day 68 boost (day 196). For all cohorts, the day 196 peak was greater than the day 84 peak. Only subjects that were positive for neutralizing antibodies against either PUUV or HTNV on the day 252 time point were asked to return for the day 365 time point. The percentage positive for that day 365 subset of subjects was based on the original number of subjects in the efficacy evaluable population for each cohort (30 subjects). Accordingly, the day 365 anti-HTNV and PUUV seropositivity rates were approximately 50% for all cohorts one year after the first vaccination (Table 3). There were no significant differences in the PsVNA_50_ neutralizing antibody response (i.e., responders versus nonresponders) based on sex, race, ethnicity or age (Appendix A). Moreover, there were no HLA alleles found to be significantly associated with the titer response after correction for multiple comparisons (Appendix A).

Some subjects produced neutralizing antibodies only to PUUV, some only to HTNV, and some to both viruses. Overall, 90.8% of subjects became seropositive to either HTNV or PUUV at one or more time points. The rates were 90.0%, 96.7%, 90.0% and 86.7%, for cohorts 1–4, respectively. There were only 11 of 120 subjects (9.2%) that were negative against both viruses on all time points. The percent that were seropositive for each time point against either virus was plotted (Figure 4B). Greater than 50% of the subjects administered the DNA vaccine on day 28 were seropositive on day 56, whereas subjects in the two cohorts that received the PBS vehicle alone on day 28 required the day 56 vaccination before a >50% seroconversion rate was achieved. The difference in day 56 seropositivity rates was significant between cohort 1 and the two cohorts receiving PBS vehicle alone on day 28 (cohorts 2 and 4). The seropositivity rates went above 50% for all groups after the first series of administrations (day 84) and then remained >50% for all subsequent time points. The peak seropositivity time point was on day 196 for all cohorts.

Responses to both viruses at one or more times points were observed in 76.7%, 73.3%, 80.0% and 66.7%, for cohorts 1–4, respectively. The overall seropositivity to both viruses for all 120 subjects in the study after vaccination was 74.2%. The percent seropositive for each time point against both viruses was plotted (Figure 4C). The day 56 neutralizing antibody response was significantly higher in the cohort that received DNA vaccine on day 28 (cohorts 1 and 3). Those same two cohorts were the only two to reach 50% seropositive on day 84. Rates dropped until after the day 168 boost when all four cohorts achieved >50% seropositivity. On day 196 cohort 3 had the highest seropositive percentage (88.5%) and this was significantly higher than the cohort 4 percentage (60.0%). Subjects positive on day 252 were asked to return for the day 365 visit (Table 3). Cohorts 2 and 3 retained a ≥50% seropositivity rate (based on total number of subjects in cohort) against both viruses on day 365 and all four cohorts retained a ≥50% seropositivity rate against HTNV or PUUV.

Individual PUUV and HTNV PsVNA50 titers for each subject were plotted in Figure 2. The PsVNA50 GMT of the efficacy evaluable population for each cohort at each timepoint were calculated and plotted (Figure 5). GMT peaked after the first series of vaccinations (day 84) and then again after the day 168 boost (day 196) for both PUUV and HTNV. Although the titers dropped after day 196 they still remained greater than the preboost (day 168) titers for all four cohorts. The only significant differences between the cohorts were on day 56 and day 196 (Figure 5C). The difference between the cohorts receiving the DNA vaccine at the day 28 time point (cohorts 1 and 3) versus PBS vehicle alone (cohorts 2 and 4) on day 56 indicated that the day 28 immunization significantly increased the day 56 titers. The significant difference on day 196 was between cohort 3 and 4.

For most subjects, peak neutralizing antibody responses occurred on day 84 (after the first series of DNA vaccine administrations during the first two months of study) and then again on day 196 (after the day 168 boost). In all cohorts, the day 196 peak was greater than the day 84 peak. A breakdown of the day 196 response is illustrated in Figure 6. Cohort 3 had the highest peak day 196 GMT, median titers and seropositivity rates against HTNV (GMT = 456, median 428, seropositivity 92.3%) and PUUV (GMT = 223, median = 263, seropositivity 88.5%).

### 3.4. Comparison of Neutralizing Antibody Responses Measured by PsVNA and PRNT 

The PRNT has been used historically to evaluate the neutralizing antibody responses to hantavirus vaccines [12,13,24,25]. Day 84 specimens were evaluated by HTNV and PUUV PRNT. HTNV and PUUV PRNT_50_ GMT for Day 84 sera showed no statistical differences among the four cohorts (Appendix A). The PUUV PRNT50 GMT for all cohorts were approximately two-fold greater than the HTNV PRNT50 GMT (Table 4 and Appendix A). The percentage of seropositive samples for PUUV PRNT50 and median titers were also greater for all groups.

To compare the data obtained by PsVNA with PRNT, the day 84 results of the two assays were analyzed using four different methods: Pearson product moment correlation, mountain plot, Bland–Altman plot and Deming regression analysis. Results from the Pearson product moment correlation of log-transformed HTNV PRNT50 and log-transformed HTNV PsVNA50 showed a strong correlation (r = 0.83, *p* < 0.01; Figure 7A). Results from the Pearson product moment correlation of log-transformed PUUV PRNT50 and log-transformed PUUV PsVNA50 also showed a moderate correlation (r = 0.70, *p* < 0.01; Figure 7B). Mountain plots, or folded empirical cumulative distribution plots, are created by computing a percentile for each ranked difference between methods. To get a folded plot, the following transformation was performed for all percentiles above 50 percentile = 100 percentile. These percentiles were then plotted against the differences between the two methods. Symmetrical plots centered over 0 suggest that the two methods were unbiased with respect to each other. Unsymmetrical plots with long tails suggest differences between methods. For HTNV, the mountain plot was centered near zero, but the left tail was larger and longer than the right, suggesting that PRNT_50_ values were smaller than the respective PsVNA50 values (Figure 7C). For PUUV, the mountain plot was centered near zero, but the right tail was larger and longer than the left, suggesting that PRNT50 values were larger than the respective PsVNA50 values (Figure 7D). Bland–Altman plots show the relationship among the differences between a pair of measurements and the average of a pair of measurements. For each subject, the difference between PRNT_50_ and PsVNA_50_ was calculated and the average of the PRNT50 and the PsVNA50 was calculated. These were then plotted against each other. Ideally, 95% of data points fell within 1.96 standard deviations of the mean difference. The data points also should show no discernable pattern of systematic differences. For the HTNV graph, 93.5% of data points fell within 1.96 standard deviations of the mean difference (Figure 7E). For the PUUV graph, 92.7% of data points fell within 1.96 standard deviations of the mean difference (Figure 7F). For both HTNV and PUUV, the Bland–Altman graphs show no discernable systematic differences. Deming regression analysis assesses the presence of a constant bias and proportional bias between two methods using a regression model that adjusts for error on both the x- and y-axes. Constant bias refers to a difference between methods that remains constant over measurement pairs. Proportion bias refers to a difference between methods that varies according to the magnitude of the measurement. The results of the Deming regression model are shown in Figure 7G. If the confidence interval for the intercept contains zero, then it can be concluded that a constant bias does not exist between the two methods. If the confidence interval for the slope contains 1, then it can be concluded that a proportional bias does not exist between the two methods. The model suggests that neither proportional bias nor constant bias exist between assays for HTNV. However, the model also suggests that both proportional bias and constant bias exist between assays for PUUV (i.e., PUUV PRNT50 titers were consistently higher than the respective titers PsVNA50).

### 3.5. HLA Results

Enrolled subjects HLA were determined. Logistic regression analysis results for the association of each allele at a locus with the PsVNA50 titer response (responder or nonresponder) are summarized in Appendix A. No alleles were found to be significantly associated with the titer response after correction for multiple comparisons. Given that the sample size of 130 subjects was relatively small for HLA analysis, these results were not surprising. There were several alleles that were found to be significant prior to multiple comparison correction, and these alleles may be of future interest when examining the association of PUUV and HTNV response with alleles in a larger study.

Before correction for multiple comparisons, several alleles were significantly associated with the PUUV PsVNA50 response. Subjects expressing the DRB1*03 allele had a lower probability of response than those who expressed all other DRB1 alleles combined (OR = 0.37, 95% CI: 0.16–0.83; *p* = 0.0167, q = 0.4447). Subjects expressing the DRB1*13 allele had a higher probability of response than those who expressed all other DRB1 alleles combined (OR = 2.67, 95%CI: 1.05–6.75; *p* = 0.0385, q = 0.4447). Subjects expressing the G*01*01 allele had a higher probability of response than those who expressed all other G*01 alleles combined (OR = 2.13, 95% CI: 1.00–4.51; *p* = 0.0491, q = 0.4447). 

Before correction for multiple comparisons, several alleles were significantly associated with HTNV PsVNA50 response. Subjects expressing the A*02 allele had a lower probability of response than those who expressed all other A alleles combined (OR = 0.45, 95% CI: 0.22–0.91; *p* = 0.0260, q = 0.4047). Subjects expressing the B*15 allele had a lower probability of response than those who expressed all other B alleles combined (OR = 0.29, 95%CI: 0.12-0.72; *p* = 0.0076, q = 0.2425). Subjects expressing the DRB1*03 allele also had a lower probability of response than those who expressed all other DRB1 alleles combined (OR = 0.43, 95% CI: 0.19–0.997; *p* = 0.0493, q = 0.4047).

## 4. Discussion

This study includes the first use of a gene optimized HTNV DNA vaccine in humans. In preclinical and clinical studies with an earlier, non-optimized version of the HTNV DNA vaccine, pWRG/HTN-M(x), we observed a reduced anti-HTNV neutralizing antibody response when the HTNV and PUUV plasmids were combined, as opposed to delivered separately [12,13,15,26]. In contrast, in this study we found that the gene optimized HTNV DNA vaccine plasmid, pWRG/HTN-M(co), successfully elicited high levels of anti-HTNV neutralizing antibodies when delivered in combination with the PUUV DNA vaccine plasmid, pWRG/PUU-M (s2). Although we cannot rule out the possibility that modest levels of interference or enhancement occurs with multivalent immunization compared to responses achievable with monovalent HTNV or PUUV administration, the results of the present study indicate that the previously observed interference (i.e., reduced anti-HTNV neutralizing antibody responses when the combined HTNV and PUUV vaccine was administered) can be mitigated through refinements in the DNA vaccine design.

Only 9% of subjects did not respond to either vaccine at any time point. The root causes for nonresponse in these subjects remains unknown. There was no detectable correlation between nonresponders with sex, age, race or ethnicity. Similarly there were no HLA alleles found to be significantly associated with the titer response after correction for multiple comparisons. Given that the sample size of 130 subjects was relatively small for the HLA analysis, these results were not surprising. There were several alleles that were found to be significant prior to the multiple comparison correction, and these alleles may be of future interest when examining the association of PUUV and HTNV response with alleles in a larger study.

An objective of this study was to determine if reducing the initial series of vaccinations from three (days 0, 28 and 56) to two (days 0 and 56) would result in lower seropositivity or GMT. There was a significant (*p* < 0.05) reduction in the day 56 seropositivity rates and GMT when the day 28 vaccination was excluded; however, significant differences were not observed at any subsequent time points. Nevertheless, there was a trend of lower seropositivity and GMT for the two groups (cohorts 2 and 4) that did not receive the day 28 vaccination. This was the case for both the anti-PUUV and anti-HTNV responses. We concluded that it would be possible to reduce the initial series of vaccinations from three to two; however, the level of antibodies would be lower, especially during the second and third month after the initial vaccination series. For travelers, where there might be a need to elicit a more rapid immune response before traveling to an endemic area, including the Day 28 vaccination would be beneficial (e.g., cohort 1 or cohort 3).

Another objective of this study was to determine if a reduction in the dose of DNA from 2 to 1 mg per vaccination would affect vaccine immunogenicity. There were no significant (*p* <0.05) differences in seropositivity or GMT between cohort 1 and 3, or between 2 and 4 indicating that reducing the dose by half did not have a major effect on immunogenicity. Interestingly, cohort 3 (1 mg, four vaccinations) had the highest seropositivity rate and highest GMT at the most time points. This same dose of vaccine given as three vaccinations (cohort 4) exhibited the lowest seropositivity for the later time points.

The day 168 booster vaccination clearly resulted in an increase in the neutralizing antibody responses regardless of the initial vaccination series (e.g., two or three vaccinations) with peak GMT detected one month after the boost (day 196) for all four cohorts. The day 196 peak GMTs were higher than the original day 84 GMT for both viruses indicating that the 6 month boost not only maintained levels of neutralizing antibodies, but actually increased the magnitude of the response. After the peak response on day 196 the level of the neutralizing antibody declined rapidly in the next month and then appeared to level out over the next 4–5 months. The GMT on day 365 was similar to the GMT after the initial series (day 84; Figure 5). We speculated that individuals who respond to the vaccine could remain protected even if antibodies drop below detectable levels, due to a rapid recall immune response. Here we saw evidence of a recall response after the 6 month boost. This is in agreement with previous nonhuman primate studies using related hantavirus DNA vaccines where we observed a rapid increase in neutralizing antibodies within days following a boost more than two years after the initial vaccination [10,27].

Overall, the results of this study indicate that high frequency immune responses can be achieved across a range of doses and immunization schedules. The optimal immunization regimen for a given indication will likely depend on the specific intended use, the target population and the economic/logistical factors. For example, the cost of the drug substance (DNA) can have a major impact on the feasibility of fielding a vaccine and therefore the amount of DNA used to immunize should be minimized, but the total number of immunizations required to achieve protection is also a critical logistical factor is another important consideration. In the present study, the total amount of DNA used to vaccinate each cohort was reduced by 2 mg between each cohort (see Table 5) and, in general, the response declines as the dose declines. The exception to this trend is cohort 3. Despite using a total of only 4 mg of DNA for the immunization regime, the neutralizing antibody response for cohort 3 was more robust than the cohorts using 6 mg (cohort 2) and 8 mg (cohort 1) of DNA (Table 5). However, when the 1 mg dose was used, then the full four-vaccination regime was necessary because if the day 28 vaccination was skipped (i.e., as in cohort 4) then the neutralizing antibody response dropped to 80.0–66.7% (Table 5). 

Two types of assays to measure neutralizing antibody responses to the vaccines were used in this study. All specimens were tested in the PsVNA assay, and a subset of these (day 84) was assayed by PRNT. In previous comparisons of the PsVNA and PRNT, including human specimens from vaccine studies, we had found no discernible differences between the PsVNA and PRNT in any of the analyses performed (unpublished data). In those studies the number of specimens was relatively low (i.e., <50). Here, the day 84 data pairs from 123 subjects were used. In three of the four analyses we found the assays were statistically similar. The Pearson correlation indicated that significant linearity exists between the methods for both HTNV and PUUV. The mountain plots suggested that a slight bias might be present for the PUUV assays (i.e., several specimens had high anti-PUUV PRNT titers but undetectable PUUV PsVNA titers). The Deming regression results suggested an absence of bias between the PRNT assay and the PsVNA assay for HTNV; however there was a constant bias and proportional bias between the PRNT assay and the PsVNA assay for PUUV. Specifically, the PUUV PRNT appeared to detect neutralizing activity that was missed by the PsVNA. The PsVNA exclusively measures the capacity of antibodies to prevent virus entry into cells, whereas the PRNT measures prevention of entry as well as inhibition of later steps in the virus replication processes including egress and spread. PUUV plaques/foci are more difficult to detect than HTNV plaques/foci; thus, it is possible that antibodies that do not prevent entry but do interfere with virus spread might have a more consequential effect on the PUUV than the HTNV plaques/foci even when highly diluted in the semisolid overlay. If this were the case, then the apparent increased sensitivity of the PUUV PRNT could explain the assay bias. Another possibility is that the PUUV and HTNV DNA vaccines are eliciting cross-neutralizing antibody responses that are contributing to the overall neutralizing activity against HTNV and PUUV, respectively. If the PsVNA and PRNT differentially detect such cross-neutralizing activity differently, then a bias might be observed. For example, if the HTNV DNA vaccine elicited cross-neutralizing antibodies that prevented PUUV plaque formation, but not entry, then that could explain the subtle difference between the PUUV PRNT and PsVNA. It will be of interest to analyze larger numbers of specimens from subjects vaccinated with individual HTNV or PUUV DNA vaccines, rather than the combination vaccine, to further investigate the observed bias in the PUUV assays.

## 5. Conclusions

We have shown that a two-component DNA vaccine for HFRS administered in healthy adult volunteers as a 1:1 mixture using the Ichor Medical Systems IM-EP delivery device exhibited an unremarkable safety profile while inducing high frequency immune responses against the target antigens. Our studies provide a strong rationale for further large scale testing of these hantavirus vaccines to advance them toward licensure.

## Figures and Tables

**Figure 1 vaccines-08-00377-f001:**
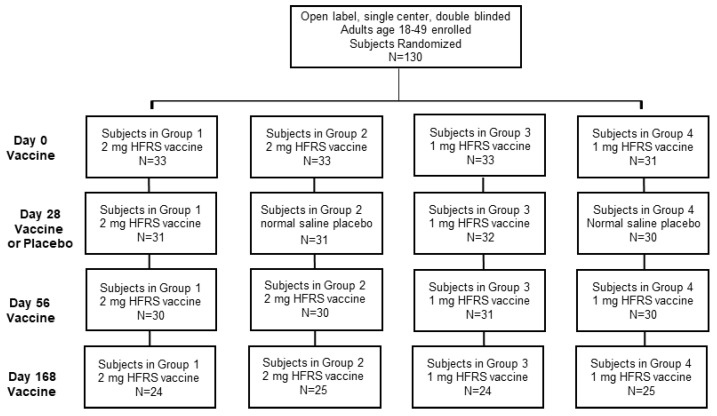
Clinical trial.

**Figure 2 vaccines-08-00377-f002:**
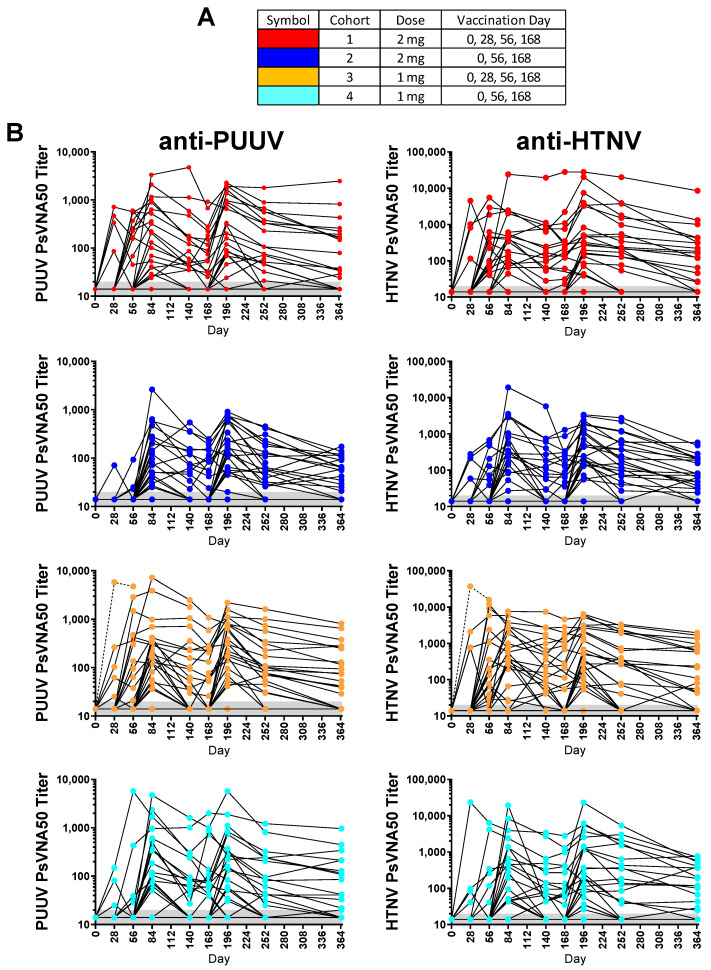
Pseudovirion neutralization assays (PsVNA)50 titers. Puumala virus (PUUV) and Hantaan virus (HTNV) PsVNA50 titers for each subject at each timepoint grouped by cohort. Subjects were randomized and vaccinated with the indicated amount of DNA (either 2 mg or 1 mg) on the indicated day (0, 28, 56, 168 or 0, 56 and 168). (**A**) Legend. (**B**) PUUV and HTNV PsVNA50 titers. Note that the Y-axis for PUUV and HTNV graphs are different. The limit of quantitation was a PsVNA50 titer of 20 (grey shaded area). Dashed line indicates specimens that were not included in the efficacy evaluable population.

**Figure 3 vaccines-08-00377-f003:**
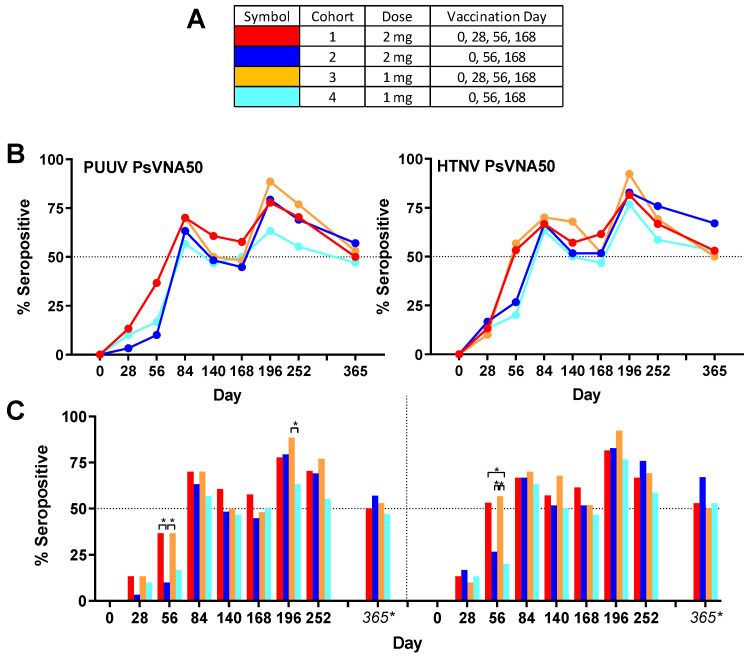
Anti-PUUV and anti-HTNV PsVNA50 seropositivity rates. (**A**) Legend. (**B**) PUUV and HTNV seropositivity rates for each cohort presented as line graphs. Percentages are based on the number of subject presenting non-missing data. (**C**) The same data plotted as bar graphs to highlight significant differences between groups. Percentages are based on the number of subject presenting non-missing data except Day 365. The only subjects who returned for Day 365 were the subpopulation that was still seropositive on the Day 252 visit. For Day 365 the percentage is based on the full cohort size of 30. * *p*-values from pairwise two-sided Fisher’s exact tests.

**Figure 4 vaccines-08-00377-f004:**
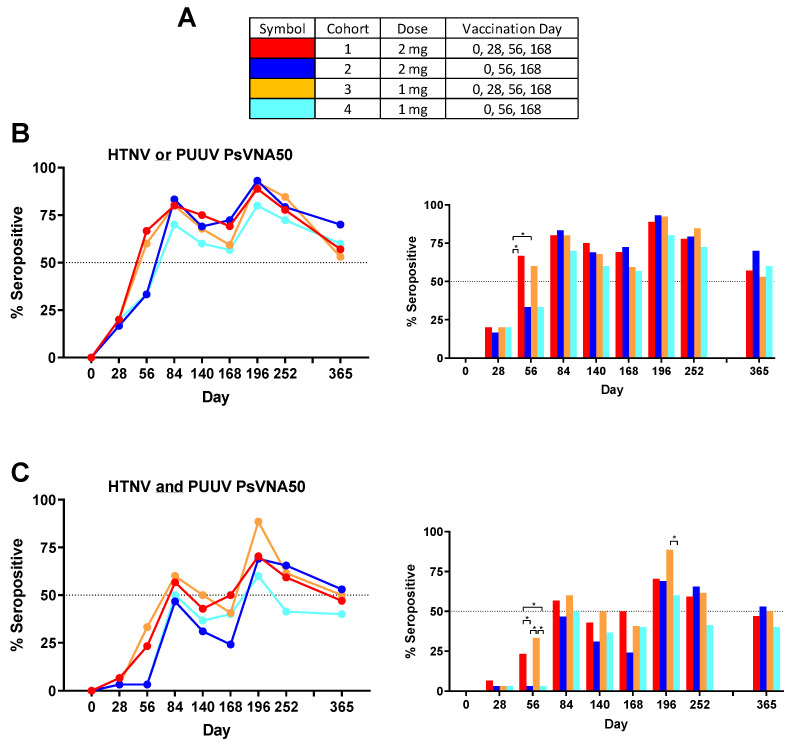
Seropositive to either HTNV or PUUV; or to both viruses. (**A**) Legend. (**B**) The percentage of subjects seropositive against either HTNV or PUUV at each timepoint were plotted as line graphs and bar graphs. (**C**) The percentage of subjects seropositive for both PUUV and HTNV at each timepoint was plotted. Percentages are based on the number of subject presenting non-missing data except day 365. The only subjects who returned for day 365 were the subpopulation that was still seropositive on the day 252 visit. * *p*-values from pairwise two-sided Fisher’s exact tests.

**Figure 5 vaccines-08-00377-f005:**
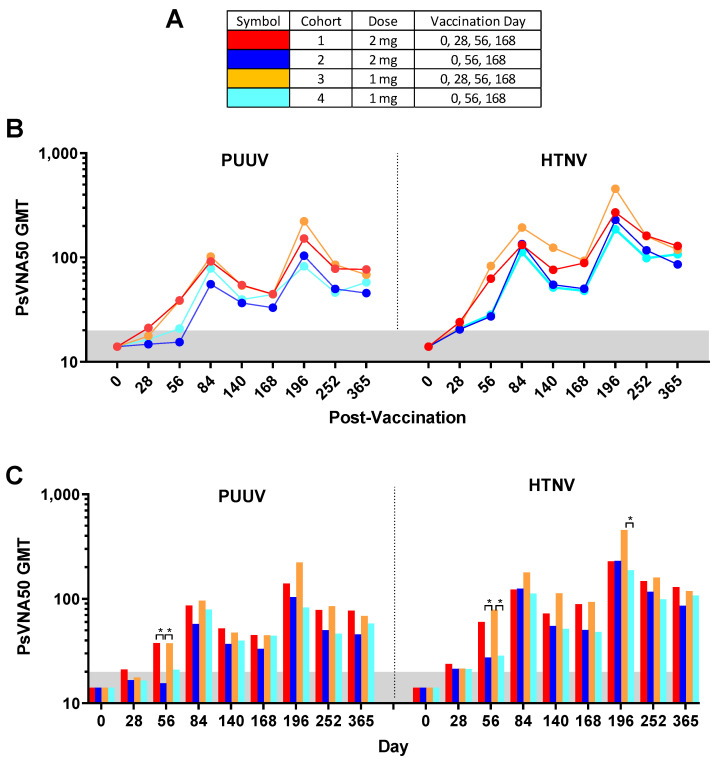
PsVNA geometric mean titers. (**A**) Legend. The PsVNA50 GMT for PUUV and HTNV presented as line graph (**B**) or bar graph (**C**). The limit of quantitation was 20 (grey area). Values <20 were given a value of 14.1. * *p*-value from post-ANOVA pairwise comparison based on the analysis of log10-transformed titers.

**Figure 6 vaccines-08-00377-f006:**
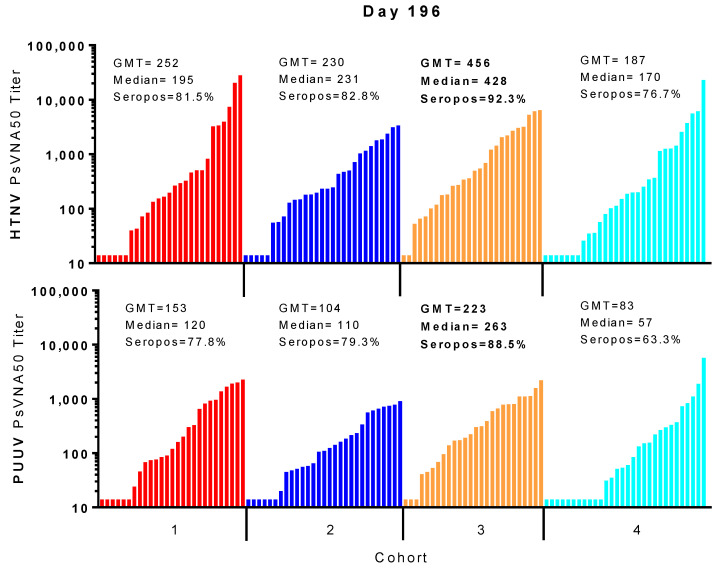
Peak PsVNA geometric mean titers (Day 196). Cohort geometric mean titers peaked on day 196. Cohorts are the same as defined in Figure 1A. Individual PsVNA50 titers were sorted lowest to highest and plotted for HTNV (top panel) and PUUV (bottom panel). GMT, median and % seropositive are plotted. Highest values are highlighted in bold text.

**Figure 7 vaccines-08-00377-f007:**
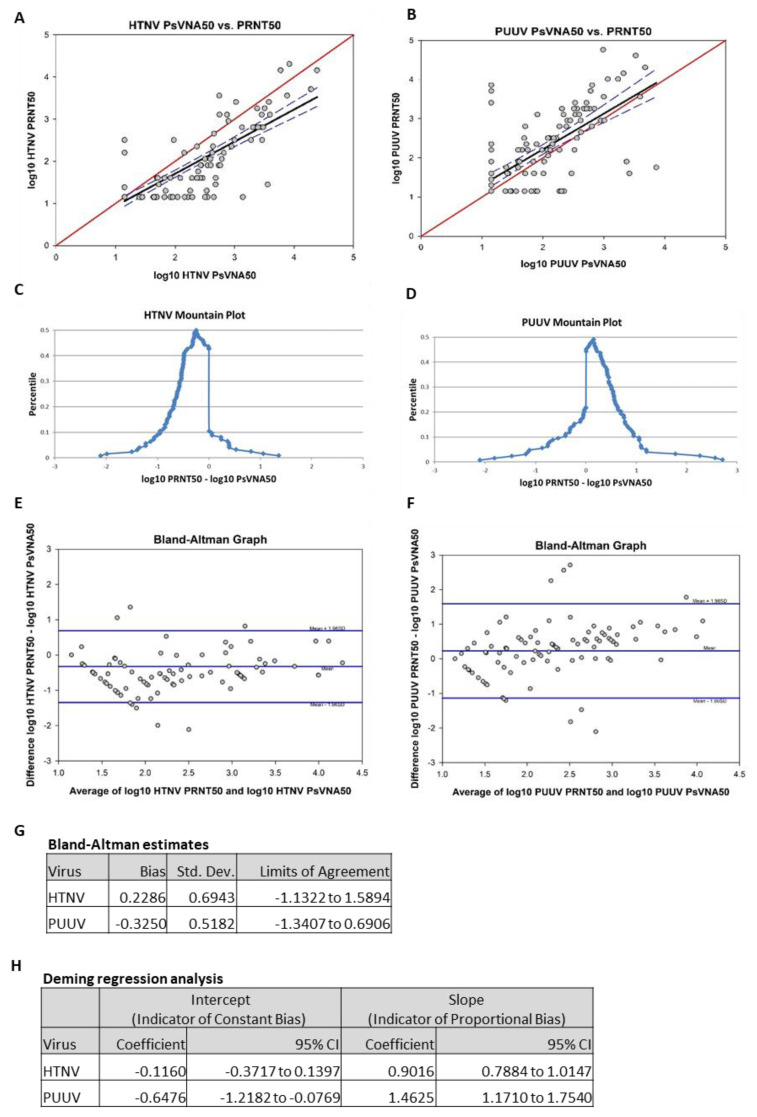
(**A**,**B**) In each graph, the black points represent the PRNT50-PsVNA50 pairs. The red line represents the ideal linear function of perfect correlation. The black line represents the observed linear function corresponding to the r value, and the blue lines represent the 95% confidence interval around the linear function. (**C**,**D**) In each graph, the blue dots represent the difference between PRNT50 and PsVNA50 for each subject plotted against each observation’s percentile value. (**E**,**F**) In each graph, the black dots represent the difference between PRNT50 and PsVNA50 plotted against the average of the PRNT50 and the PsVNA50 for each subject. Blue lines represent the mean bias and limits of agreement. (**G**) The bias estimates and limits of agreement from the Bland–Altman analysis are shown. (**H**) The results of the Deming regression model are shown. If the confidence interval for the intercept contains 0, then it can be concluded that a constant bias does not exist between the two methods. If the confidence interval for the slope contains 1, then it can be concluded that a proportional bias does not exist between the two methods.

**Table 1 vaccines-08-00377-t001:** Study demographics.

Number of Subjects	Total Number	% of Total
Enrolled	130	100
Completed Study Procedures	120	92
Withdrawn/replaced	10	8
Total	130	100
**Sex**	**Total Number**	**% of Total**
Male	67	52
Female	63	48
Total	130	100
**Age**	**Total Number**	**% of Total**
18–19 years	2	1
20–29	64	49
30–39	37	29
40–49	27	21
Total	130	100
**Race**	**Total Number**	**% of Total**
African American	64	49
Asian	4	3
Other	9	10
White	53	39
Total	130	100
**Ethnicity**	**Total Number**	**% of Total**
Hispanic/Latino	11	8
Not Hispanic/Latino	119	92
Total	130	100

**Table 2 vaccines-08-00377-t002:** Solicited adverse events for each cohort.

Adverse Event and Severity	HTNV/PUUV 2.0 mg Three Dose (*n* = 33) ^a^	HTNV/PUUV 2.0 mg Two Dose (*n* = 33)	HTNV/PUUV 1.0 mg Three Dose (*n* = 33)	HTNV/PUUV 1.0 mg Two Dose (*n* = 31)
Erythema at Injection Site Grade 1 (Mild)	6 (18.2%)	6 (18.2%)	3 (9.1%)	3 (9.7%)
	6 (18.2%)	6 (18.2%)	3 (9.1%)	3 (9.7%)
Bruising at Injection Site Grade 1 (Mild) Grade 2 (Moderate)	3 (9.1%)	1 (3.0%)	2 (6.1%)	3 (9.7%)
	2 (6.1%)	1 (3.0%)	2 (6.1%)	3 (9.7%)
	1 (3.0%)	0	0	0
Swelling at Injection Site Grade 1 (Mild) Grade 2 (Moderate)	2 (6.1%)	0	1 (3.0%)	1 (3.2%)
	1 (3.0%)	0	1 (3.0%)	0
	1 (3.0%)	0	0	1 (3.2%)
Pain at Injection Site Grade 1 (Mild) Grade 2 (Moderate) Grade 3 (Severe)	33 (100%)	30 (90.9%)	33 (100%)	29 (93.5%)
	24 (72.7%)	20 (60.6%)	27 (81.8%)	24 (77.4%)
	9 (27.3%)	9 (27.3%)	5 (12.2%)	5 (16.1%)
	0	1 (3.0%)	1 (3.0%)	0
Myalgia Grade 1 (Mild) Grade 2 (Moderate) Grade 3 (Severe)	8 (24.2%)	8 (24.2%)	4 (12.1%)	3 (9.7%)
	6 (18.2%)	6 (18.2%)	4 (12.1%)	2 (6.5%)
	1 (3.0%)	2 (6.1%)	0	1 (3.2%)
	1 (3.0%)	0	0	0
Muscle Contractions Grade 1 (Mild) Grade 3 (Severe)	4 (12.1%)	1 (3.0%)	1 (3.0%)	1 (3.2%)
	3 (9.1%)	1 (3.0%)	1 (3.0%)	1 (3.2%)
	1 (3.0%)	0	0	0
Fatigue Grade 1 (Mild) Grade 2 (Moderate) Grade 3 (Severe)	12 (36.4%)	13 (39.4%)	4 (12.1%)	9 (29.0%)
	9 (27.3%)	6 (18.2%)	3 (9.1%)	8 (25.8%)
	3 (9.1%)	7 (21.2%)	0	1 (3.2%)
	0	0	1 (3.0%)	0
Headache Grade 1 (Mild) Grade 2 (Moderate) Grade 3 (Severe)	8 (24.2%)	9 (27.3%)	7 (21.2%)	7 (22.6%)
	5 (15.2%)	8 (24.2%)	5 (15.2%)	4 (12.9%)
	3 (9.1%)	1 (3.0%)	1 (3.0%)	3 (9.7%)
	0	0	1 (3.0%)	0
Lymphadenopathy Grade 1 (Mild)	3 (9.1%)	4 (12.1%)	3 (9.1%)	0
	3 (9.1%)	4 (12.1%)	3 (9.1%)	0
Axillary Pain Grade 1 (Mild) Grade 3 (Severe)	4 (12.1%)	4 (12.1%)	2 (6.1%)	2 (6.5%)
	3 (9.1%)	4 (12.1%)	2 (6.1%)	2 (6.5%)
	1 (3.0%)	0	0	0
Tachypnea/Increased Respiratory Rate Grade 1 (Mild)	0	3 (9.1%)	0	2 (6.5%)
	0	3 (9.1%)	0	2 (6.5%)

^a^ One subject was withdrawn due to non-study related SAE (hypoglycemia). >25% cells are shaded.

**Table 3 vaccines-08-00377-t003:** Durability of the neutralizing antibody response.

Cohort	# of Subjects Returning for Day 365 Blood Collection	PUUV Seropositive *n* (% of Total)	HTNV Seropositive *n* (% of total)	PUUV or HTNV Seropositive *n* (% of Total)	PUUV and HTNV Seropositive *n* (% of Total)
1	20	15 (75.0%)	16 (80%)	17 (85%)	14 (70.0%)
2	22	17 (77.3%)	20 (90.9%)	21 (95.5%)	16 (72.7%)
3	22	16 (72.7%)	15 (68.2%)	16 (72.7%)	15 (68.2%)
4	20	14 (70.0%)	16 (80%)	18 (90.0%0	12 (60.0%)

Seropositive is defined by a PsVNA50 titer ≥ 20.

**Table 4 vaccines-08-00377-t004:** Day 84 plaque reduction neutralization test (PRNT)50

HTNV			PUUV	
Cohort	GMT	% Positive	GMT	% Positive
1 (*n* = 31)	50.5	48	138.2	68
2 (*n* = 31)	54.6	52	90.4	68
3 (*n* = 31)	81.7	52	144.6	68
4 (*n* = 30)	70.0	57	173.3	67

**Table 5 vaccines-08-00377-t005:** Dose vs. neutralizing antibody response (PsVNA50).

Cohort	Dose	Doses	Total DNA	% Seropositive HTNV and PUUV	Day 196 HTNV GMT	Day 196 HTNV Median	Day 196 PUUV GMT	Day 196 PUUV Median
1	2 mg	4	8 mg	76.7%	252.4	195.0	152.8	120.0
2	2 mg	3	6 mg	73.3%	230.3	231.0	104.2	110.0
3	1 mg	4	4 mg	**80.0%**	**456.3**	**428.0**	**222.9**	**262.5**
4	1 mg	3	3 mg	66.7%	187.0	169.5	82.8	57.0

Highest response among cohorts shown in bold. Seropositive is a PsVNA50 ≥ 20; mg = milligram; GMT = geometric mean titer.

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
