# Peer review of "A Phase 2a Randomized, Double-Blind, Dose-Optimizing Study to Evaluate the Immunogenicity and Safety of a Bivalent DNA Vaccine for Hemorrhagic Fever with Renal Syndrome Delivered by Intramuscular Electroporation"

_vaccines, 2020, doi:10.3390/vaccines8030377_

Round 1

Reviewer 1 Report

Dose optimizing study for the bivalent DNA vaccine done by the authors can be accepted after minor clarifications.

1) Dosage 2mg has given 50 % more injection site issue than 1mg, but the seropositive response is similar after 84 days. Is 1mg dosage preferable than higher dose of bivalent vaccine ?

2) what was the site of injection of the DNA vaccine? was there any difference in the site of injection in the subjects .

3)whats the reason behind not having the placebo or control comparing to single vaccine with bivalent vaccine.

4) 28th day vaccine looks essential since 4mg total dosage gives highest 80 % response. Irrespective of the ethinicity , travelling condition is this dosage and frequency mandatory ? 

Reviewer 2 Report

This manuscript by Hooper et al presents data from a phase 2a clinical trial evaluating the immunogenic potential and safety of a bivalent DNA vaccine against Hantaan virus (HTNV) and Puumala virus (PUUV), two of the major causative agents of hemorrhagic fever with renal syndrome (HFRS). Major goals of this study were to evaluate the optimized bivalent DNA vaccines for the effects of vaccine dose and frequency of intramuscular electroporation administrations on the magnitude and durability of neutralizing antibody titers. The clinical trial is well-designed, the manuscript is well-written and data are presented clearly and support the conclusions authors have reached. I have a few concerns that needs to be addressed:

  1. The differences in the neutralization titers of the PUUV PRNT assay vs PsVNA is rather puzzling. Given the timeline of 10 days of the PUUV PRNT assay, it is unlikely that the antibodies are still blocking the egress and spread of virus as the authors have surmised in line 450-51. Did the authors include sera in the overlay and supplement the overlay with more sera containing overlay periodically? Inclusion of some pictures of the HTNV and PUUV plaques could be informative here especially as the authors state that PUUV plaques are harder to detect. It is much more likely that the neutralization titers in PUUV PRNT is overestimated.
  2. What is the basis of choosing PsVNA50 titer of 20 as responder? Is a PsVNA50 titer of 20 protective against acquisition of HTNV or PUUV infection in animal models?
  3. Can the authors include a couple of sentences to explain how exactly the mountain plots in Fig 7C-D were generated? For example, how were the percentiles (Y-axes) were calculated?
  4. Fig 7 - The labels are too small to be legible. Please update. Match the axis scale labels in 7A and 7B.
  5. Fig 7 legends, Line 427 - "H)" should be "G)"

Minor points:

  • Line 269 - "oneunanticipated" should be "one unanticipated" 
  • Table 2 - Which cohort (2?) does the footnote "a" belongs to?
